# Smart Systems: The Role of Advanced Technologies in Improving Business Quality, Performance and Supply Chain Integration

Ivett Schmidt [1], Wyn Morris [1], Andrew Thomas [1,*] and Louise Manning [2]

1    Aberystwyth Business School, Aberystwyth University, Aberystwyth SY23 3DL, UK; ivs3@aber.ac.uk (I.S.);
     dmm@aber.ac.uk (W.M.)
2    Lincoln Institute for Agri-Food Technology, University of Lincoln, Lincoln LN2 2LG, UK;
     lmanning@lincoln.ac.uk
*    Correspondence: ant42@aber.ac.uk; Tel.: +44-01970622725

**Abstract:** Technologies such as the Internet of Things (IoT), cloud computing, and Smart Systems (SSs) have become an important focus for industry, especially in the manufacturing and retail sectors. The aim of this paper is to analyse the importance of SSs and their related technologies in improving business performance. Through an initial systematic review of sixty-one papers, the authors identify six key determinants that lead to the effective application of SSs in business systems, namely, the application of effective managerial skills, supply chain integration, financial analysis, business performance, strategic and operational capabilities, and technologies. The work then goes on to develop an industry case study that informs thinking on the capabilities of smart technologies in collaborative working environments and then onto the development of a practice-focused framework for future research. This study identifies that the implementation of SSs within organisations not only improves business performance but also their supply chains through the effective integration of business activities and systems, and through the degree to which communication and decision-making is facilitated between humans and devices. This leads to the improvement of quality, speed of information, and information sharing. This study also finds that there is a lack of systems standards that currently govern IoT and SSs integration and data security within businesses.

**Keywords:** smart; systems; quality improvement; supply chain; integration; collaborative systems; smart technologies

## 1. Introduction

Smart Systems (SSs) and their associated digital technologies have transformed business models. The spread of information and communications technology (ICT) has brought many benefits to businesses in commercial marketing, finance, and customer service with fast and accurate communication supporting improved decision making, product development, quality of product, and service often measured through increased customer satisfaction. ICT includes electronic tools and systems ranging from electronic databases, e-mail, through to e-business systems [1]. Furthermore, ICT is defined as 'all electronically mediated information exchanges, both within an organisation and with external stakeholders supporting the range of business processes' in both buying and selling [2]. The concept of e-business focuses on the online communication between firms, suppliers, and customers, which supports the business performance, hence, the speed of the supply chain. Companies recognise that they must not only implement, but continuously invest in e-business technologies, to integrate processes and systems in their supply chain [3].

A supply chain is considered here as the movement of products, information, money, and the flow of knowledge [4]. Organisations need to manage a controlled supply chain to improve sales, profit, and competitiveness [5]. Supply chain transparency through creating

greater visibility from effective information sharing between customers and suppliers is critical in these operating environments [6]. SSs supply the connection between supply-chain actors and any value-adding activity, especially in the production planning process where the 'quality' of information required is critical to the future productive effectiveness of a business. Further, meeting increasing market demand and accelerating innovation requires a more sophisticated organisational approach to information sharing within the wider context of supply-chain management [7].

The Internet of Things (IoT) is considered the next generation of Internet-connected embedded ICT systems in a digital environment and establishes and manages the networked connection between physical devices (e.g., machines, vehicles) and human beings [8]. The IoT gathers and shares data on Internet-based global platforms generating real-time databases [9]. This data collection function is supported by a rising number of potential technologies [10]. Different applications help effective communication processes such as wireless sensor networks (WSN), radio frequency identification (RFID), and global positioning systems (GPS). Technologies can control, for example, the required level of stock in a distributed manufacturing system, which leads to improved product quality which in turn leads to higher customer satisfaction and profitability [8]. The IoT has the capability to improve the quality of supply chain processes by increasing; visibility, accuracy, traceability, and collaborative decisions [11] through the three viewpoints [10] being, things-oriented, internet-orientated, and semantic-oriented.

Since the 20th century, many technological developments (such as the Internet) have spread globally undoubtedly bringing improvements and changing business models [12] and practices associated with timely service and the range of products offered. Organisations, and wider supply networks, require this innovation to meet continued and changing consumer needs in a timely way [7]. New business models are required as the Internet removes the physical and time barriers along the supply chain between customers and sellers and firms and their suppliers. Furthermore, trading partners benefit from the dynamic capabilities of e-business in terms of their cost structure, the development of collaborative value networks, and the information-sharing aspects afforded by internet connectivity [12]. This enhanced dynamic capability creates the ability for e-businesses to change their business models with some technologies making a great impact on the online retail sector. Indeed, an agile alliance of networked machines and systems improves manufacturing operations, production quality, and supply chain capabilities [13], and flexibility within operations, providing innovative solutions, and improved performance while value is created [5,7]. In this respect the supply chain has become data-driven, optimising traceability, and promoting smart manufacturing.

McKinsey [14] defines the four main dimensions of IoT, which offer benefits for the production system. They are connectivity, speed of view, accessibility, and anchoring. Connectivity and accessibility reflect the relevance of real-time data and up-to-date production information. Speed of view addresses day-to-day operations, which intend to meet demand as fast as they can, changing production processes within the organisation and potentially altering the length of the supply chain. With advanced technology solutions and the proliferation of e-commerce, organisations recognise the efficiency of a shorter supply chain. Thus, the establishment of close cooperation with local suppliers may have a positive effect on businesses' productivity and market success. However, when utilising IoT, Hill [3] highlights that the organisation should do 'things right' rather than 'the right things'. This shows the move towards IoT playing an increasing role in the 'management' of business functions going forward where the quality of data and information is critical for accurate decision making. Figure 1 shows how IoT technologies enable business objectives to be achieved through systems that provide fast flexible connectivity, and flow of parts and information. These capabilities encourage efficiencies within a smart manufacturing system, which improves further efficiency of the supply chain with a 'cycling effect' on each other.

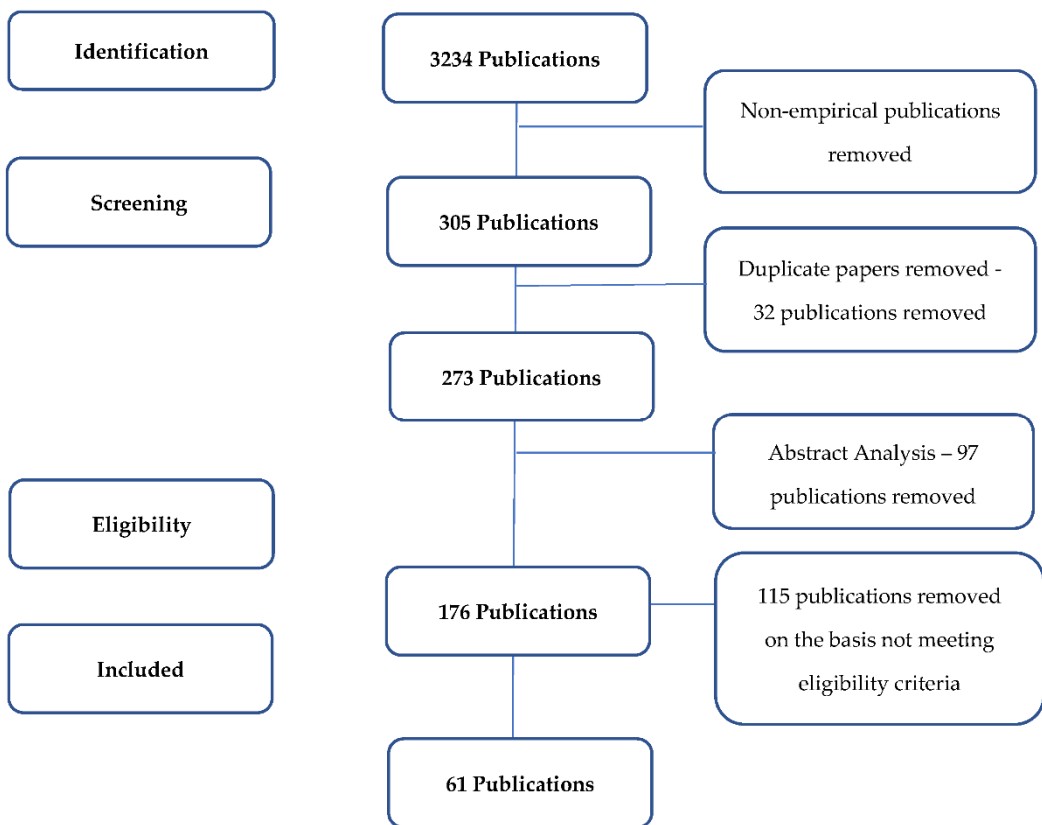

**Figure 1.** The flow chart of the systematic review methodology.

SSs use technology for establishing and maintaining communication with all supply chain partners, improving the production process, and shaping how manufacturing information systems and associated practices influence supply chain performance [15]. Akkermans et al. [16] reinforce the need to use competitive information and IT applications in business relationships within the supply chain. The streamlining of data and the uninterrupted interaction with a connected environment result in a smooth operational and distribution performance if partners can access real-time information through this electronic connection with each other [17].

The e-procurement process in particular affects the efficiency of the supply chain in terms of inventory levels, deliveries, and customer satisfaction [18]. SSs can reduce demand uncertainty and inaccuracy and demand risk related to supply visibility. If organisations collaborate, coordinate, and cooperate to share real-time data about customer demand, transportation costs, location, and level of inventory, through the information technologies and systems, the supply chain can be visible to every party and be more effective [15]. In 2019, Amazon had one of the biggest web services infrastructures of any organisation. Its cloud computing managed a significant volume of data that was generated by SSs [19]. Cloud computing is the online platform where a user can store data to access it remotely and easily [20]. The ability to store data and then be able to access and analyse it easily underpins decision-making and supply chain innovation. As PWC (2020) states, most companies believe digital solutions support an increase in efficiency and can improve relationships with local suppliers. In 2016, it was suggested that $178 billion had been invested in SSs in manufacturing whilst in comparison $78 billion had been invested in transportation [21].

According to Deloitte [22], hospitals became the leading investors in 2020 in SSs, as a result enhancing social aspects of their activities and trust with patients. Additionally, 29% of the respondents in PWC's survey [21] have implemented networking technologies and 60% of them plan to invest in them by 2022. The interest in SSs is expected to grow further, not only because of the higher revenue it could generate and the benefits of

process tracking, but also because of the need for online business solutions driven by the coronavirus epidemic [23]. Investment in SSs in manufacturing was $27.76 billion in 2018 and is predicted to reach $ 136.83 billion by 2026 [24].

The purpose of this study is to consider the role of SSs in improving business collaboration, decision-making, and supply chain integration. Four research questions are considered in this research:

RQ 1.: What impact does IoT have on e-business?
RQ 2.: How does IoT influence the quality and speed of information flow in the supply chain in the production environment?
RQ 3.: How can companies measure the effective implementation of IoT in their supply chain?
RQ 4.: Is IoT a business management tool that can accelerate innovation and change in a business model thereby driving organisational processes to deliver improved performance?

An initial systematic review of sixty-one research publications identified six key thematic areas that were seen as key determinants in SSs being able to deliver impact in a business. The review assists in being able to explain the nature and type of impact that businesses can obtain from the application of SSs. The review also allows for the development of a practice-focused framework that assists in answering the remainder of the research questions. The paper then uses an industry scenario to critically analyse the synthesized data, and identify, then reflect on the impact of IoT and SSs on a business supply chain. In summary, the application of SSs has gained ground quickly in almost every industry and is evolving rapidly. SSs are leading a reimagining of advanced manufacturing technology, involving sensors, industrial robots, cameras, and other technologies [25]. Thus, their development is of interest here where the main scientific results identified in this paper outline the personal contribution made to the scientific literature by the authors.

## 2. Materials and Methods

In order to conduct the study and explore the potential findings, a multi-method approach was adopted that involves a systematic review of literature, the development of a practice-focused framework, and subsequent critique through application in an industry example provided in the form of a case study [26]. The results are presented in a narrative form [27] to enable reflection on the framework developed. This process of systematically combining literature and practice-informed data is supported by extant studies that have used the combination of the theory and empirical observations [28]. The focus of the systematic review is to identify the scope and gaps in existing knowledge, to then inform the second phase of the research and provide insight into the research questions [26].

*Systematic Literature Review Design*

The systematic literature review was based on a keyword search using Elsevier Science Direct (http://www.sciencedirect.com/ (accessed on 21 June 2022)); Emerald, (http://www.emeraldinsight.com/ (accessed on 21 June 2022)); and IEEE Explore https://ieeexplore.ieee.org/Xplore/home.jsp (accessed on 21 June 2022)) databases. Seven search queries were applied to each of the three key databases. The search terms included 'IoT', 'Smart Systems', 'Internet of Things', 'IoT AND Smart Systems', 'Smart Manufacturing Systems' 'digital supply chain', and 'e-commerce'. The initial output from the keyword search yielded a total of 3234 publications. Search string matches were sought in publication titles, author-supplied keywords, or abstracts. A second stage 'sifting' was applied to remove papers that were nonempirical. This left 305 publications to review. Of the 305 publications, these were further reduced to 273 publications by removing duplicate publications. Through further screening of the titles, abstracts, and conclusions of the remaining papers for relevancy against the search terms, the remaining 273 papers were reduced to 176 publications. The 176 publications were subsequently read in full and accepted or rejected depending on whether they focused on aspects such as productivity, the efficiency of the supply chain, financial performance, and the challenges related to the

advanced technology in the company. This left 61 papers that were identified as relevant to this research (Figure 1 and Table A1). The sources were then thematically analysed to form the theoretical framework that is presented in the next section of the paper. The order of the papers in Table A1 is systematically derived and shows the number of characteristics highlighted by each paper.

The second element of the methodology was to then consider an industry scenario that allowed the theoretic framework to be critiqued to provide a practice-informed framework. The industry scenario was considered using a case study methodology that enables the investigation of a real-life phenomenon in its complexity [28]. Case study research operates with the elements of existing data collection, scenario observation, and combining multimedia or oral histories to provide triangulation through the analysis and synthesis of different sources [29]. Creswell [30] distinguished between three types of case studies, explanatory, exploratory and descriptive perspectives. This study used a descriptive research design for the analysis of the scientific papers and articles [31]. It also presented a comprehensive description of the importance of advanced technologies in e-business. Various secondary data sources were used such as press releases, media and journal articles, books, interviews, and company websites [26].

The sources were considered for both their validity and their value in contributing to the narrative. The advantage of this methodology is that it allows data to be iteratively drawn together from a range of sources. Further, applying both quantitative and qualitative analyses of the data to address the research questions counteracts the weaknesses in both types of data when used alone. The qualitative aspect helps to explain the complexities of real-life cases in context [32] and the quantitative method aspect supports objectivity and generalisability [33,34] in real life. The scenario business utilised in this research was Amazon. In this case, the scenario business has been chosen to reflect a given situation and the application of SSs in that specific situation, rather than a case study that is representative of the industry sector. The business model is critiqued through comparison with organizations such as Walmart, Siemens, and others.

## 3. Results

### 3.1. Introduction

The literature derived from the systematic review was read and thematically analysed which led to the identification of six key determinants. The sixty-one papers are listed in Table A1. The six determinants were identified as:

- Adequacy of managerial skills;
- Efficiency of the supply chain i.e., cooperation between the organisation and suppliers;
- Financial aspects, such as reduced costs, or increased profit;
- Performance of business i.e., productivity, quality of product, and service;
- Strategic or operational challenges related to technology such as privacy, security, and lack of skills;
- Technology use such as RFID.

A number of texts focussed on the use of SS technologies in manufacturing and logistics and supply chain systems [35,36]. Some sources review cloud computing and IoT and the associated benefits [37], and the development of stronger partnership between firms and their suppliers [38,39]. The literature verifies that IoT improves business performance and operation, thus, it improves supply chain efficiency through cooperation with suppliers. However, there is a lack of informative business models around the development of SSs [40] and challenges with employing the right professionals also hinder the realisation of the potential opportunities associated with SSs, proving a barrier to innovation [41].

The next section evaluates the application of SSs in a specific scenario with the example of Amazon and the use of advanced technology to primarily improve the efficiency of their supply chain i.e., their performance and cooperation with suppliers.

*3.2. Business Scenario: Amazon*

Amazon is one of the most relevant real-world examples to explore the research questions posed in this research. Amazon is a Seattle-based online e-commerce retailer that operates internationally. It offers a wide range of different products and services and has developed a technological infrastructure and application software for businesses and individuals on its website. The company is striving for a multi-legged position which strengthens its diversity in the market [42]. The importance of Amazon's applied warehousing technology is interesting, and the business model is based on advanced technology, but mostly on cloud computing in which Microsoft and IBM are the other two leading market players [43].

Amazon has over 175 fulfillment centres covering multiple destinations worldwide such as North America, Europe, and Asia [44]. One of the primary objectives of its fulfillment centres is to meet current demand and to reflect an understanding of customers' needs through data analysis. In addition, they offer advanced logistics and operational support to suppliers including tracking systems, inventory management (monitoring and controlling stock and the picking process) delivery information, and customer service in their warehouses. SSs applications facilitate a 'from start to finish 'fulfilment process [44]. Investing in technology builds capacity and capability helping to maximise production through automation, faster shipping times, optimum inventory levels, and reducing costs supporting the price level provided for customers [45]. The acceptable balance between the role of humans and automation is difficult, with vulnerabilities in both investing in technology and automation or in funding upskilling and engaging staff. For example, Amazon have in recent years experienced multiple staff strikes in their warehouses around the world [46].

Amazon's geographic positioning supports faster fulfillment which is a key driver of Amazon's business model [47] with the business benefit derived from their global distribution network. In comparison, Walmart, one of Amazon's biggest rivals, has many physical stores in several varieties of formats worldwide including supercentres, and discount stores. Walmart can reach more customers physically and build stronger trust with its return policy, which allows customers to return items to any store, providing more service flexibility. However, this strength may have been weakened during the coronavirus pandemic when there was a switch to online purchasing. One pre-Covid survey, suggests customers prefer to order their desired products online and pick them up at Walmart stores rather than ordering them online and waiting for them at home [48]. Further, Walmart was one of the first companies who adopted RFID to track inventory [49].

Amazon has established a digital ecosystem for its suppliers to improve their performance in the market. This approach can support collaboration between Amazon and its partners in the supply chain making the business environment more sensitive, adaptable, and aware of any value chain inefficiency [50]. Amazon uses SSs with IoT linked to cloud-based applications and its advanced algorithm to analyse its partners rather than integrating other systems into its own system. Amazon uses its system in each process for inbound and outbound logistics, concentrating on improving timeliness and saving money with automation, machine learning, and artificial intelligence (AI) [51].

As a comparison, Siemens uses AWS platforms which have led to an 85% reduction in its costs [51]. Siemens has built its own applications based on AWS services which create an ecosystem partnership between them in production and operations. Therefore, Siemens suggests it can improve its productivity through the development of its 'online order-to-delivery collaboration platform' (Siemens, n.d.), the consolidated operations, and incremental offerings to its customers [52]. Another example of the cost efficiency of advanced technology is Intuit, which is a business and financial software company. It uses AWS to reduce its operational costs by 25% [53]. Shell also confirms this cost-saving effect, but it emphasised the real-time monitoring feature of AWS solutions, which ensures flexibility and scalability for the company [54]. Another remarkable case is Pentair, which provides water filtration systems to breweries and fish farms. Pentair increased its

productivity by 10% due to its collaboration with AWS, while it also reduced production costs. This manufacturer utilises sensors to gather data on filtration systems with AWS IoT applications and it enables them to analyse this data remotely on an AWS cloud service [55].

Pentair can collect data about the quality of beer in the breweries or monitor the environmental conditions in fish farms through the use of appropriate sensors. The AWS cloud platform can be used to store these data. Users can analyse the data anytime and anywhere to provide solutions rapidly. This system improves business performance through a high level of collaboration with partners connecting machines with custom web portals through to its users (operations, managers and suppliers) [56–60].

Amazon intended to launch a drone-delivery solution in the US market in 2020 [61]. This opportunity creates a threat, as e-commerce is seen by new entrants as attractive and requiring low capital for entry, however, there are potential issues with Internet interruption and cyber-attacks [62]. Cloud-based solutions enable an organisation to associate and interconnect products, machines, and systems. This requires Amazon to analyse, evaluate and share the data collected by IoT with their partners, thus achieving greater collaboration with other actors in the supply chain and making their cost–benefit balance more favourable [63].

Amazon focuses on its own business performance i.e., productivity through warehousing technology and digital ecosystem partnerships. They also concentrate on supply chain efficiency with their suppliers. They improve the speed of information flow with SSs in the supply chain. This provides a cost-efficient solution in inventory and logistics management. These examples demonstrate that adopting advanced technology provides a more sophisticated process for companies to increase productivity and reduce their operational costs concurring with published literature sources considered in this study. The research questions are now considered in sequence to structure the results section.

RQ1.: What impact does IoT make on e-business?

Internet-based IoT business solutions have revolutionised relationships not only for business to customers (B2C) but also for business to business (B2B) [64]. Organizations concentrate on two particular business interests: increasing customers' satisfaction and therefore profit and to reduce costs in inventory, procurement processes, and delivery. IoT applications such as RFID tracking systems, EDI, and cloud computing provide an effective solution for businesses to manage these interests.

Amazon exemplifies how a company can take advantage of IoT in its manufacturing and logistics system and become one of the leading global e-retailers. Amazon did not seek to categorise all items in its inventory, rather it used the advanced technology for tracking and tracing them improving the preparatory packing and delivery processes. Its innovative robotic solutions support the previously mentioned business interests. The literature stresses the benefits of real-time reaction and the cost-reducing effect of IoT [65,66]. This means that IoT can make a great impact on e-retailers' productivity and capability underpinned by both theoretical and practical facts.

RQ2.: How does IoT influence the quality and speed of information flow in the supply chain in the production environment?

As shown inter alia [66] in the literature review, the real-time information, and SSs support the flexibility and transparency of the supply chain from a theoretical viewpoint. IoT can shorten the length of the supply chain improving companies' productivity and reducing costs which are the most important benefits for companies as highlighted in RQ1. SSs can support the decision-making process via information sharing both internally and externally to the organisation and through the entire supply chain. SSs can establish an impressive collaboration among partners, however, most companies increase customers' satisfaction primarily with a fast and adequate reaction to demand using the advanced technology. This research has found that few articles emphasise the creation of a shorter supply chain as one of the major reasons for using IoT or other advanced technologies related to it in an SS. Most articles mention this beneficial effect in terms of an economical

side effect rather than a specific focus. For example, according to one of the McKinsey surveys [14], the first three expectations in using SSs are fewer lost sales, lower inventory, and lower operational costs. After that companies expect an optimising network from IoT solutions [67]. Further on, the biggest benefit of SSs adoption is noticed in improved data management and analysis [68].

Compared with the above-mentioned studies and reports, Amazon positions its successful business model on the cloud computing service, and the exchange and transparency of supported information promoted by the technology. Therefore, one of Amazon's businesses is succeeding primarily through SSs by optimising the supply chain. The digital ecosystem partnerships of Amazon with Pentair and Siemens support the theoretical framework that IoT applications can shorten procurement processes and reduce supply chain complexity.

RQ3.: How can companies measure the effective implementation of IoT in their supply chain?

Implementation of IoT may be monitored and evaluated indirectly rather than directly. Therefore, through investing in SSs solutions, companies can estimate the impact on their profitability and productivity especially by reducing the cost of inventory and the time to purchase items. In this context, the positive impact of IoT investment on the profitability and efficiency not only of the company but of the entire supply chain can be expressed as an increase in customer satisfaction. Customer satisfaction is closely related to the speed and accuracy of the delivery of ordered goods, and the development of a favourable price, which can be cascaded through to consumers. Here again, this study highlights that IoT applications provide, both optimisation of inventory costs, and the transparency of the supply chain through the exchange of up-to-date real-time information.

Amazon takes advantage of this benefit by implementing IoT use in warehousing technology in its fulfillment centres worldwide which supports maximising production and reducing operational costs. Further, it reduces delivery time thus, Amazon can simplify the logistics process with adequate tracking systems and information sharing with other participants in the supply chain.

RQ4.: Is IoT a business management tool that can accelerate innovation and change in a business model thereby driving organisational processes to deliver better performances?

This question considers factors such as the level of IT and technology development within the company. Many SSs applications can forecast aspects of business processes and improve business efficiency [69,70]. Therefore, SSs are a driver for increasing productivity and embedding the IoT, through SSs, is conducive to business success.

SSs solutions make a positive financial and operational impact on the business model by improving operations and efficiency [71]. Amazon is a great illustration of how to use a multi-layered business model for achieving a higher level of productivity. Through innovation, they focus on effective organisational processes based on time, cost, and information flow to deliver better performance. For example, when referring to the concise comparison of AWS and Azure, Amazon holds the leading position overall despite Microsoft performing better in some aspects. Amazon connects itself to offer services and products and then develops them based on user feedback. Thus, Amazon can take advantage of its economic scale. Although companies should evaluate the existing conditions for implementing SSs, after the implementation of IoT, they should adopt certain organisational aspects such as data and inventory management within organisations. Neither the reviewed literature nor the Amazon case study emphasizes strongly that implementing and effectively using SSs requires adequate managerial skills.

Some studies mention that implementing and effectively using SSs requires adequate managerial skills. In contrast, Amazon case studies do not emphasize the importance of managerial skills and their development. Additionally, the sources examined do not highlight that Amazon's complex business model claims that using IoT as a business management tool improves the existing business model.

## 4. Discussion

The systematic review compared extant literature with a real case example in this study. The literature review highlighted six determinants that have been positioned in a theoretical framework (Table A1) to inform the analysis of the Amazon case study. The approach highlighted similarities between theory and practice and the gaps between them. The case study provides a reflection on the importance of SSs within supply chains and shows that companies can increase their quality levels and business performance with the effective application of IoT within SSs. The consequence of renewed associations using SSs is the shortening of the supply chain due to the enhanced information sharing, thus, making the supply chain more agile in its reaction to demand. A more accurate measurement system to assess the degree of efficacy of the implementation of IoT would be useful but there are currently no standards for integration nor processes to determine the degree of efficacy. A study by [72] identified a similar issue of integrating innovation systems with quality management systems. Their study identified a high degree of commonality between innovation and quality-systems standards which could allow for future integration of the two areas. The current gap between IoT integration with formal quality management systems should be addressed in future research and industry activity.

Financial benefits such as reducing inventory costs have been highlighted herein, but concerns have also been raised around privacy and data sharing issues and the vulnerability to a cyber-attack. This research has shown that SSs are a notable driver of productivity but human factors such as the appropriate management resources and leadership style and appropriate workforce skills frame the degree of benefit derived. The company needs an effective leader who is able and willing to manage the changes that result from the implementation of SSs.

The implementation of SSs is a complex solution, which influences not only e-procurement processes but also innate aspects of the business model. The example of Amazon indicated that IoT solutions are a key tool for the management and delivery of timely organisational processes, thus ensuring a productive business model. Amazon sets an example of how IoT can accelerate the speed of information in the supply chain thus, improving cooperation with suppliers. These key features are also underlined by the result of the systematic review in this study. Amazon's partners such as Siemens and Pentair point out the positive impact of the IoT on their productivity and costs. Hereby, the real case example (Amazon) supports the literature review.

Neither Amazon nor its partners propose any standard measurement process which can directly measure the activity of the implemented advanced technology in the supply chain. Additionally, none of them mention, in the sources considered, that effective managerial skills are crucial to implement and take advantage of the IoT. Specifically, it is not highlighted by Amazon that the SSs require the existing leadership style or vice versa.

Consequently, the research study provides valuable evidence on the importance and benefits of SSs while highlighting practical issues in its adoption and application. Further, this research identified the gaps between the theoretical and the practical viewpoints with the systematic review. The main challenges in implementation and effective usage of the IoT are to identify and manage the data sharing and privacy issues and to appraise the degree of the implemented IoT. The important gap is to apply an accurate leadership style to manage the challenges and accomplish the integration of the advanced technology within the external and internal business process. The findings highlight that there are some data and privacy security issues associated with SSs applications that must be considered by organisations to reduce their vulnerability. These standards are currently absent worldwide. There is also a need for further investigation into the required managerial skills needed to match the specific IoT structure developed with the right business model.

In summary, the six determinants derived from the literature can be combined into a practice-informed framework namely: adequacy of managerial skills, promoting supply-chain efficiency, providing financial return through reducing costs and increasing profitability, improving performance through higher productivity, and the use of technology,

addressing strategic operational challenges such as technology scale-up [73], as well as data privacy, security, and a lack of workforce knowledge and adequacy of managerial skills also need to be considered. These 6 determinants within the 4 research questions present the advantages of IoT, such as a more effective performance in the supply chain and the developed cooperation with suppliers.

## 5. Conclusions

The objective of this study was to analyse the importance of SSs and their related technologies towards improving business performance. The systematic review of sixty-one papers identified six determinants that were considered key to the effective application of SSs in business systems namely, the application of effective managerial skills, supply chain integration, financial analysis, business performance, strategic and operational capabilities, and technologies.

This study identified that the implementation of SSs within organisations not only improves business performance but their supply chains too through the effective integration of business activities and systems and through the degree to which communication and decision-making are being facilitated between humans and devices. This leads to the improvement of quality, speed of information, and information sharing. The study also found that there is a lack of systems standards that currently govern IoT and SSs integration and data security within businesses.

The main scientific results identified in this paper outline the personal contribution made to the scientific literature by the authors. The work found that the implementation of SSs and IoT improves information accuracy and information sharing between humans and devices, as well as companies and their suppliers. Thus, SSs ease the decision-making process associated with information sharing through effective communication which also leads to better delivery of consumer satisfaction, hence customer quality. IoT applications support a more effective supply chain, a notable driver for productivity. Additionally, it can reinforce management performance within the supply chain. An adequate leadership style could increase the efficiency of the implemented SSs, but few studies focus on managerial skills. The other challenge is data security. Improving the practical implementation of SSs requires improved governance of data security, verification activities, and the application of supply chain standards for SSs by policymakers.

The principal limitation of this study is that due to the constraints of the COVID-19 pandemic there is a lack of primary data. Whilst the study would have been better informed with the inclusion of primary data in terms of reliability and validity, the findings address the research questions, suggest opportunities for improvements in the sector, and inform future research in this area. Another limitation is that the secondary data available focused mainly on B2C rather than B2B relationships. It would be beneficial in future research to analyse B2B interactions within SSs and more specifically, to consider in more depth the connection between the level of adoption of advanced technology and the operational efficiency within the supply chain from which it is derived.

**Author Contributions:** Conceptualization: I.S. and W.M.; methodology, L.M.; software, I.S.; validation, A.T.; formal analysis, I.S., W.M. and L.M.; investigation, I.S.; resources, A.T.; data curation, I.S. and L.M.; writing—original draft preparation, I.S.; writing—review and editing, A.T.; visualization, W.M.; supervision, W.M., A.T. and L.M.; project administration, I.S.; funding acquisition, A.T. All authors have read and agreed to the published version of the manuscript.

**Funding:** This research received no external funding.

**Institutional Review Board Statement:** Not applicable.

**Informed Consent Statement:** Not applicable.

**Data Availability Statement:** Not applicable. The study did not involve primary data from participants.

**Conflicts of Interest:** The authors declare no conflict of interest.

**Appendix A. List of Selected Journal Articles Following Systematic Review and Screening**

The key for the six determinants is:

- Adequacy of managerial skills;
- Efficiency of supply chain i.e., cooperation between the organisation and suppliers;
- Financial aspects such as reduced costs, or increased profit;
- Performance of business i.e., productivity, quality;
- Strategic or operational challenges related to technology such as privacy, security, and lack of skills;
- Technology use such as RFID.

**Table A1.** Analysis of Literature on Smart Systems and the identification of the six key determinants.

| Authors | Title | Adequacy of Managerial Skills | Efficiency of Supply Chain | Financial Aspects | Performance of Business | Strategic or Operational Challenges | Technology Use | Number of Determinants Identified |
|---|---|---|---|---|---|---|---|---|
| [5] | Using the Internet of Things in a production planning context | | X | | X | | | 2 |
| [6] | eBusiness and supply chain integration | | X | | | | | 1 |
| [7] | Using the Internet of Things in a production planning context | | X | | X | | | 2 |
| [12] | A multi-domain trust management model for supporting RFID applications of IoT | X | | | X | | | 2 |
| [18] | Exploring new technologies in procurement | | X | | X | | | 2 |
| [34] | Toward Industry 4.0 with IoT: optimizing business processes in an evolving manufacturing factory | | | | | | X | 1 |
| [35] | IoT-based production logistics and supply chain system—Part 2. IoT-based cyber-physical system: a framework and evaluation. | | | | X | X | X | 3 |
| [36] | IoT-based production logistics and supply chain system—Part 1. Modeling IoT-based manufacturing supply chain. IoT aware logistics Systems, could computing and agriculture | | X | | X | | | 2 |
| [37] | The future of retail supply chains | | X | | | | | 1 |
| [38] | Radically rethink your strategy: How digital B2B ecosystems can help traditional manufacturers create and protect value | | | X | X | | | 2 |
| [39] | The future of the Internet of Things: toward heterarchical ecosystems and service business models. | | X | | X | | | 2 |
| [40] | Guide to IoT innovation (SME focus) Achieving innovation performance | | | | X | | X | 2 |
| [42] | Digital economy report 2019. Value creation and capture: implications for developing countries. | | | X | X | | | 2 |

**Table A1.** *Cont.*

| Authors | Title | Adequacy of Managerial Skills | Efficiency of Supply Chain | Financial Aspects | Performance of Business | Strategic or Operational Challenges | Technology Use | Number of Determinants Identified |
|---|---|---|---|---|---|---|---|---|
| [46] | Supply chain inventory collaborative management and information sharing mechanism based on cloud computing and 5G Internet of Things | | X | | | | X | 2 |
| [48] | Coronavirus: IoT in challenging times | | X | | X | | | 2 |
| [49] | Review of RFID and IoT integration in supply chain management | | | | | | X | 1 |
| [56] | Internet of Things (IoT): Security Challenges, Business Opportunities & Reference Architecture for e-commerce | | X | | X | | | 2 |
| [59] | Smart factory performance and Industry 4.0 | | X | | | X | X | 3 |
| [63] | The Internet of Things (IoT): Applications, investments, and challenges for enterprises. | | | X | | | | 1 |
| [64] | Value co-creation practices in business-to-business platform ecosystems | | X | | | X | | 2 |
| [65] | Smart e-commerce systems: current status and research challenges. | | X | | X | | | 2 |
| [66] | Supply Chain 4.0—the next-generation digital supply chain, consumer goods | | X | X | X | | | 3 |
| [67] | The IoT business index 2020: a step change in adoption | | X | | | | | 1 |
| [68] | Improving business process and functionality using IoT based E3-value business model. | | X | | | | | 1 |
| [69] | Reinventing workflows. Power your digital transformation and drive greater impact by modernizing processes | | X | | | X | X | 3 |
| [70] | Taking the pulse of enterprise IoT | | | | X | | | 1 |
| [74] | The Internet of Things and the Modern Supply Chain | | X | | X | | X | 2 |
| [75] | The future of retail supply chains | | X | | | | | 1 |
| [76] | Long and short supply chain co-existence in the agricultural food market on different scales: | | | | | X | | 1 |

**Table A1.** *Cont.*

| Authors | Title | Adequacy of Managerial Skills | Efficiency of Supply Chain | Financial Aspects | Performance of Business | Strategic or Operational Challenges | Technology Use | Number of Determinants Identified |
|---|---|---|---|---|---|---|---|---|
| [77] | Food oligopolies, local economies and the degree of liberalisation of the global market. | | | | | X | | 1 |
| [78] | Improving business process and functionality using IoT based E3-value business model. | | X | | | | | 1 |
| [79] | Industries leading IoT revolution | | X | | | X | | 2 |
| [80] | Knowledge and skills of industrial employees and managerial staff for the Industry 4-0 implementation | X | | | | | | 1 |
| [81] | Internet of Things and its impact on business analytics | X | | | X | | | 2 |
| [82] | Interdisciplinarily exploring the most potential IoT technology determinants in the Omnichannel e-commerce purchasing decision-making processes | X | | X | | X | | 3 |
| [83] | A case study: IoT in logistics and supply chain management: evaluating the adoption rate, associated challenges and impact on cost and business efficiency | X | | X | X | X | | 4 |
| [84] | The roles of internet of things technology in enabling servitized business models: A systematic literature review | X | | X | X | X | | 4 |
| [85] | From intelligent manufacturing to smart manufacturing for industry 4.0 driven by next generation artificial intelligence and further on. | X | X | X | X | X | | 5 |
| [86] | Internet of Things: vision, application areas and research challenges | | X | | | | | 1 |
| [87] | Circular dairy supply chain management through Internet of Things—enabled technologies | | X | | X | | | 2 |
| [88] | Big data analysis of IoT-based supply chain management considering FMCG industries | | X | | X | | | 2 |

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
