# Peer review of "Smart Systems: The Role of Advanced Technologies in Improving Business Quality, Performance and Supply Chain Integration"

_standards, doi:10.3390/standards2030020_

Round 1

Reviewer 1 Report

The authors of "Smart Systems: The role of advanced technologies in improving business quality, performance and supply chain integration" present a relevant topic, namely "the importance of smart systems (SS) and technologies… to improve business performance", a relevant aspect both in the context of technological innovations and digitization of business sectors, but especially as mentioned by the authors of the paper connecting the academic area to the application business area, which makes the work have the effect of multiplying the globalization process.

The bibliographic sources, citations, key concepts used in the study are appropriately mentioned and presented by the authors of the research. Specifically, such as “providing innovative solutions and improved performance while value is created [5, 7]. In this respect the supply chain has become data driven, optimizing traceability, and promoting smart manufacturing. ” which emphasizes the orientation of the paper towards the application area.

The research methodology is based on a "multi-method approach involving a systematic review of the literature", as well as a "systematic review of the literature based on a keyword search using Science Direct, Emerald and IEEE". Moreover, the applied orientation of the methodology, respectively the analysis of the business model of the Amazon Company and its comparison with organizations such as Walmart, Siemens and others, supports from a methodological point of view the results of the paper.

The results of the research are presented by the authors of the research based on the Amazon case study, mentioned in the paper, respectively the authors point out that the analysis "does not reveal data security and privacy issues implementing and using IoT in their organization and supply chains however, the literature highlights some of them ”. However, we suggest that the authors of the research highlight the main scientific results as a personal contribution to the scientific literature, given that the results are adequately presented but very much focused on the application side of the case study.

The conclusions are presented by the authors of the research, respectively the authors emphasize for example "the importance of smart systems (SS) and related technologies" and the fact that "IoT implementation improves the accuracy of information and the exchange of information between people and devices." At the same time, the authors clearly present the limitations of the study and the future research they want to continue. However, as we mentioned in the results chapter, we appreciate that the personal scientific results that contribute to the scientific literature should be highlighted.

We congratulate the research team, we suggest the revision of the paper according to the above mentioned, and after the revision we propose for acceptance the paper.

Author Response

Thank you for your valued suggestions for improving the manuscript. Please find below your feedback and our response to your suggestion.

1. We suggest that the authors of the research highlight the main scientific results as a personal contribution to the scientific literature, given that the results are adequately presented but very much focused on the application side of the case study.

2. We appreciate that the personal scientific results that contribute to the scientific literature should be highlighted.

Answer: The authors have added a section into the conclusions to state that the  scientific results were a personal contribuution by the authors.

Thank you

Reviewer 2 Report

Very interesting article. My following comments are intended to be constructive and hope they are helpful to the authors:

Should be improved abstract, it giving more concrete and important information to the reader. The basis aim, research methods, and the innovation of the paper is totally missing.

The part of introduction does not illustrate clearly the innovation of the specific study.

Research sample - authors should extend research - not enough data

Conclusions: This section should emphasis the objective and main results presented in this study. Research limitations and future lines of research need to be better developed.

Author Response

Thank you for your valued feedback and suggestions for improvement of the manuscript. Below are the suggestions alongside our answers to how we have addressed your feedback. I hope that these adjustmnets meet with your approval.

Thank you

  1. Should be improved abstract, it giving more concrete and important information to the reader. The basis aim, research methods, and the innovation of the paper is totally missing.

Answer: Abstract has been fully revised to show overall aim, research methods employed and the innovation that comes from teh work.

2. The part of introduction does not illustrate clearly the innovation of the specific study.

Introduction section has been revised to show teh innovation that emerges from thsi study.

3. Research sample - authors should extend research - not enough data

Revised section to show the initial 'high level' trawl yielded over 3000 publications from which 61 publications were used for the final study. Revised diagram provided that shows the wider scale research approach adopted.  .

4. Conclusions: This section should emphasis the objective and main results presented in this study. Research limitations and future lines of research need to be better developed

Revised conclusion to show the objective of the study, results, limitations and future research wiork.